# 🚀 EVOLVING ROLLOUTS: Harnessing Historical Experience for Web Agent Evolution in Reinforcement Learning

**Sinuo Wang** [1 2 †] **Piaohong Wang** [3] **Tianrui Qin** [2 ‡] **Maojia Song** [4] **Qianben Chen** [5] **Qiexiang Wang** [5]
**Gengze Zhou** [1] **Zeyu Zhang** [1] **He Zhu** [2] **Dingfeng Shi** [2] **Yutong Xie** [6] **Minghao Liu** [7] **Jiaheng Liu** [8] **Ge Zhang** [5]
**Jiawei Ma** [9] **Yuchen Eleanor Jiang** [2] **Qi Wu** [1] **Wangchunshu Zhou** [2 5]

## Abstract

Agentic reinforcement learning (RL) for web search is prohibitively expensive due to long context lengths and costly environment interactions, and this inefficiency is further exacerbated by group-based optimization, which discards learning signals from entire rollout groups with zero reward variance. In this work, we propose EVOLVING ROLLOUTS, an RL framework for web-search agents that moves beyond episodic training and distills collected rollouts into in-context guidance for future policy behavior. By extracting the reward-labeled trajectories into strategic experiences, our method augments standard parameter-space optimization with implicit context-space optimization guided by prior experience. This enables the agent to recover learning signals from zero-variance rollouts, thereby fostering co-evolution between the policy and the experience repository. EVOLVING ROLLOUTS improves sample efficiency and task performance across representative web search benchmarks, with Qwen3-8B surpassing the much larger Qwen3-30B-A3B in average performance across GAIA, xBench, and HLE, and Qwen3-4B attaining comparable results on GAIA and HLE.

---

[†]Work done during internship at OPPO Agent Team; [‡]Project advisor. [1]Australian Institute for Machine Learning, Adelaide University, Adelaide, Australia [2]OPPO, China [3]City University of Hong Kong, Hong Kong, China [4]Singapore University of Technology and Design, Singapore [5]ByteDance, China [6]Mohamed bin Zayed University of Artificial Intelligence, Abu Dhabi, United Arab Emirates [7]2077.ai, China [8]Nanjing University, Nanjing, China [9]Department of Computer Science & Institute of Digital Medicine, City University of Hong Kong, Hong Kong, China. Correspondence to: Qi Wu <qi.wu01@adelaide.edu.au>, Wangchunshu Zhou <wcszhou@outlook.com>.

*Proceedings of the 43rd International Conference on Machine Learning*, Seoul, South Korea. PMLR 306, 2026. Copyright 2026 by the author(s).

## 1. Introduction

Large Reasoning Models (LRMs) have recently demonstrated strong capabilities on complex reasoning tasks, driven in part by reinforcement learning from verifiable reasoning (RLVR) (Guo et al., 2025; Achiam et al., 2023; Yang et al., 2025a). The increasing complexity of reasoning tasks necessitates access to external knowledge sources, iterative planning, and tool interaction, leading to the use of these models as autonomous agents for multi-hop, tool-augmented tasks such as web search and deep research (Zheng et al., 2025; Yao et al., 2026; Wu et al., 2025b; Li et al., 2025c; Qin et al., 2025; Chen et al., 2026; Zhu et al., 2025; Shi et al., 2025; Zhou et al., 2023a;b; Zhu et al., 2026b). This shift from static reasoning to agentic decision making introduces substantial challenges. Agentic tasks unfold over long horizons and require sustained interaction with dynamic web environments (Li et al., 2025c; Liu et al., 2025; Li et al., 2025b; Wang et al., 2025a; Liang et al., 2025), often incurring high costs from API calls, web crawling, and other external tool execution. Reinforcement learning offers a principled framework for optimizing such behaviors under delayed rewards, yet remains highly inefficient in practice: episodes are long, rewards are sparse, and policy updates critically depend on reward variance across parallel rollouts. Current Agentic web search RL (Tao et al., 2025; Wu et al., 2025a; Li et al., 2025a; Chen et al., 2026; Li et al., 2025b) predominantly relies on binary, outcome-based rewards, which results in low sample efficiency and unstable training (Yu et al., 2025b). Specifically, when all rollouts within a group either succeed or fail, reward normalization causes the policy gradient to vanish (Zhu et al., 2026a), resulting in the removal of entire batches of costly environment interactions, even though these trajectories may contain partial reasoning processes and informative failure signals. Process reward models (PRMs) partially alleviate this issue by providing denser supervision, but they rely on costly step-level annotation (Chae et al., 2025; Cui et al., 2025) and remain largely static, making them poorly suited to adapt to new reasoning patterns that emerge during reinforcement learning. These limitations highlight the need

for learning paradigms that can extract reusable learning signals from zero-reward-variance rollouts without relying on additional annotations or fixed reward structures.

To address this limitation, we draw inspiration from recent experience-based agent learning paradigms, which emphasize treating failures as valuable sources of reusable experience (Li et al., 2025c; Chhikara et al., 2025; Wang et al., 2024; Fang et al., 2025; Zhang et al., 2025b; Tang et al., 2025; Wang et al., 2025b; Yu et al., 2025a; Zhang et al., 2025a). These approaches suggest that improving agentic reasoning by reusing historical interactions can indirectly alleviate optimization difficulties, especially when reward signals are sparse. However, existing methods (Cai et al., 2025b; Ouyang et al., 2025b; Wei et al., 2025; Yang et al., 2025b) predominantly exploit experience at test time or treat experience repositories as static artifacts, without coupling experience evolution to the reinforcement learning loop itself, leaving context-space and parameter-space optimization as disjoint stages rather than co-evolving signals.

Building on this insight, we introduce EVOLVING ROLL-OUTS, a reinforcement learning framework that treats experience itself as an object of optimization. Rather than discarding rollout groups with uniform rewards, which provide little or no direct policy-gradient signal, EVOLVING ROLLOUTS distills these inherently reward-labeled trajectories into a structured and continually evolving experience repository. This repository captures reusable behavioral knowledge, such as effective strategies, failure patterns, and task-specific decision cues, and retrieves relevant experiences as in-context guidance for future episodes. By jointly optimizing the agent policy in parameter space and the experience repository in context space, EVOLVING ROLLOUTS turns otherwise wasted interactions into persistent training signal. This co-evolutionary design enables the agent to accumulate, refine, and reuse experience across episodes, improving exploration, stabilizing optimization, and increasing sample efficiency under constrained training budgets.

Across representative web-agent benchmarks, EVOLVING ROLLOUTS consistently improves both training stability and final performance. Notably, EVOLVING ROLLOUTS enables Qwen3-8B to surpass the much larger Qwen3-30B-A3B on average across GAIA, xBench, and HLE, with Qwen3-4B matching it on two of the three benchmarks. These results demonstrate that structured experience evolution provides an effective complement to parameter updates for training capable long-horizon agents. In summary, our main contributions are as follows:

- **Inefficiency in agent RL:** Zero-variance rollout groups yield degenerate gradients and hinder optimization, with a stronger effect in small-scale models.
- **Co-evolving training framework:** Policy optimization is tightly integrated with a structured experience reposi-

tory that is continuously distilled and reused.
- **Improved stability and sample efficiency:** Extensive experiments show more stable training and stronger performance under same interaction budgets.
- **Open-source framework:** We release our training framework to support reproducibility and future research.

## 2. Related Work

### 2.1. LLM-based Web Search Agents

Recent LLM-based web search agents have evolved from static retrieval to long-horizon information-seeking with autonomous planning and tool use. Systems such as WebResearcher (Qiao et al., 2025) and WebSailor-V2 (Li et al., 2025a) leverage iterative planning and reinforcement learning to operate under dynamic, partially observable web environments. To address limited supervision, prior work emphasizes scalable data construction and training, including formalization-driven task synthesis in WebShaper (Tao et al., 2025) and curriculum-based learning in WebDancer (Wu et al., 2025a). Complementary architectural designs, such as WebWalker's Explorer–Critic framework (Wu et al., 2025c) and multimodal WebWatcher (Geng et al., 2025), further improve robustness during environment interaction. Despite these advances, most work focuses on task coverage, exploration, or data scale, while the sample efficiency of reinforcement learning remains underexplored. In particular, GRPO often drops entire rollout groups when reward variance is zero, wasting expensive interaction trajectories—including failed or partially correct attempts that still carry useful reasoning signals. This motivates training paradigms that better reuse collected rollouts and extract supervision beyond standard reward-based policy optimization.

### 2.2. Experience-driven Agentic Evolution

Limitations of task-coupled memory have spurred experience-driven agent evolution, where agents distill and reuse generalizable rules from past trajectories. Early Experience (Zhang et al., 2025b) learns from reward-free self-generated interactions, while AgentKB (Tang et al., 2025) externalizes reusable patterns via in-context retrieval for cross-domain transfer. This trend emphasizes test-time learning, benchmarked by Evo-Memory (Wei et al., 2025), and instantiated by MUSE's Plan–Execute–Reflect–Memorize loop (Yang et al., 2025b) and FLEX's evolvable in-context experience library (Cai et al., 2025b). Recent work (ReasoningBank (Ouyang et al., 2025b), AgentEvolver (Zhai et al., 2025)) further distills transferable strategies and failure lessons for autonomous improvement. However, these methods decouple context-space experience evolution from parameter-space RL, and seldom co-optimize experience extraction and experience reuse, motivating unified frameworks where experience evo-

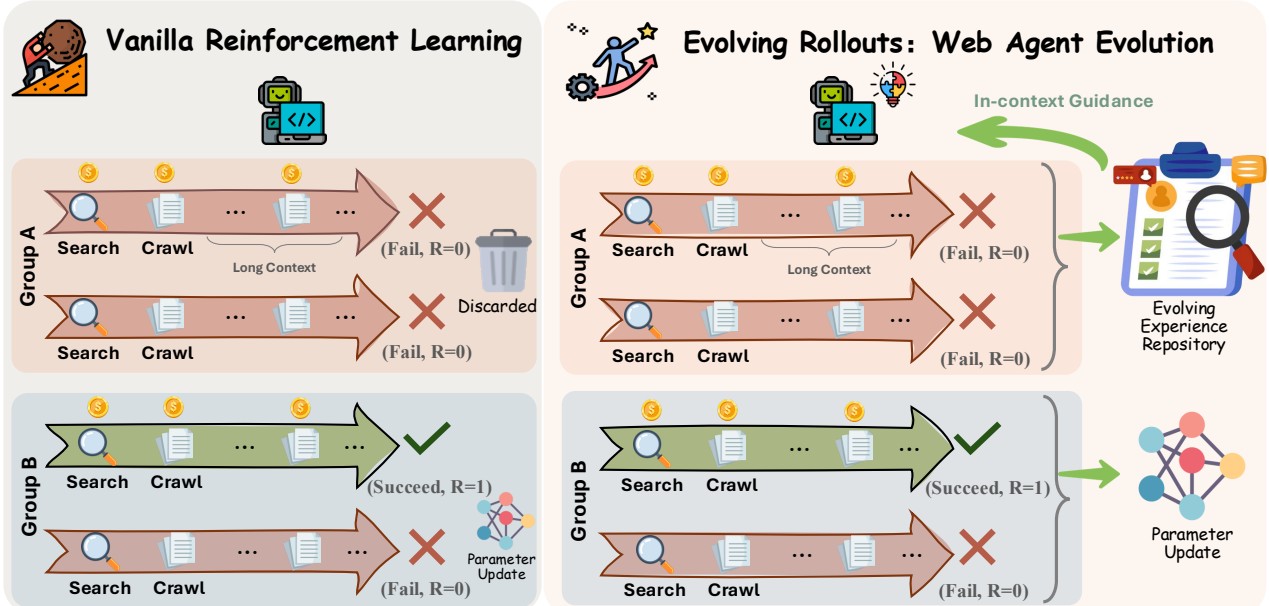

*Figure 1.* The framework of EVOLVING ROLLOUTS. Vanilla group-based reinforcement learning updates parameters only from rollout groups with non-zero reward variance (Group B), discarding signals from all costly trajectories from zero reward variance group (Group A). EVOLVING ROLLOUTS recovers the signals from zero reward variance rollout into the experience repository context space, which provides in-context guidance for future rollouts while preserving explicit parameter space optimization.

lution and RL updates co-evolve in a single loop.

## 3. Method

We present EVOLVING ROLLOUTS, a reinforcement learning framework that transforms costly web-agent rollouts into reusable and evolving strategic knowledge. Our key insight is to complement standard parameter-space optimization with a persistent context-space optimization channel, enabling experience reuse even when gradient signals vanish. Our approach operates in three stages: (1) *cold-start experience construction*, where we distill high-quality trajectories from open-source datasets into an initial experience repository; (2) *experience-driven workflow execution and supervised fine-tuning (SFT)*, where the agent retrieves relevant experiences during inference and is fine-tuned on self-generated, context-rich trajectories; and (3) *group-based RL with evolving memory*, where we apply Group Relative Policy Optimization (GRPO) on new tasks and continuously evolve the experience repository through contrastive extraction, experience consolidation, and experience updating. By co-evolving the policy and its memory, our method ensures that no successful interaction is wasted, turning every rollout into a stepping stone for future improvement.

### 3.1. Problem Formulation

We consider a web agent tasked with answering a user query $q$ through sequential interactions with a web environment.

The agent is instantiated as a Large Reasoning Model (LRM) $\pi_\theta$ that, at each time step $t$, observes an environment observation $o_t$ (e.g., webpage content) and selects an action from a discrete action space. The available actions include: SEARCH($k$), which issues a new search query $k$; CRAWL($p$), which extracts relevant information following an instruction $p$; RETRIEVE($q$), which queries the experience repository $\mathscr{E}$ to obtain previously distilled strategies relevant to the current query $q$; and ANSWER($y$), which outputs a final answer $y$ and terminates the episode.

An interaction episode induces a trajectory $\tau = (q, o_1, a_1, \ldots, o_T, a_T)$ drawn from the policy-induced distribution $\tau \sim \pi_\theta(\cdot \mid q)$. Each trajectory terminates with an ANSWER action and is assigned a scalar reward $r \in [0, 1]$ that reflects the degree of task success.

To support experience reuse, we maintain an experience repository $\mathscr{E} = \{e_1, e_2, \ldots, e_n\}$ that stores distilled, transferable reasoning strategies. Each experience $e_i$ is represented as a structured tuple:

$$e_i = (\text{title}_i, \text{desc}_i, \text{content}_i, \mathbf{v}_i, \text{success}_i) \quad (1)$$

where $\text{title}_i$ denotes a concise strategy identifier, $\text{desc}_i$ provides a one-sentence summary, $\text{content}_i$ contains detailed and transferable reasoning content, $\mathbf{v}_i \in \mathbb{R}^d$ is a $d$-dimensional semantic embedding, and $\text{success}_i$ is a binary episode-level indicator of task completion. The semantic embeddings enable efficient similarity-based retrieval and consolidation of experiences within the repository.

## 3.2. Initial Experience Repository Construction

To bootstrap the experience repository $\mathscr{E}$, we leverage publicly available web-agent trajectory datasets [1] as cold-start data. Each trajectory in these datasets is a reward-labeled rollout generated by a base agent in a web environment. We process each trajectory using a frozen Large Language Model (LLM) to extract a single, self-contained experience that distills the core reasoning strategy and action pattern leading to task success. Specifically, the LLM reformats the raw observation–action sequence into a natural language prompt–response pair suitable for supervised fine-tuning, effectively distilling each trajectory into a reusable, strategy-level learning signal. Implementation details of the extraction prompts and formatting rules are provided in Appendix A. The resulting set of distilled experiences constitutes our initial repository $\mathscr{E}$, which serves as the training data for the subsequent SFT phase.

Formally, given a trajectory $\tau = (q, o_1, a_1, \ldots, o_T, a_T)$, the extraction yields an experience instance:

$$e = \text{EXTRACT}_\phi\left(q, \{o_t, a_t\}_{t=1}^T\right) \quad (2)$$

where $\text{EXTRACT}_\phi$ denotes a frozen LLM-based extractor parameterized by $\phi$. Collecting such experiences from a set of trajectories $\{\tau^{(i)}\}_{i=1}^N$, we construct the initial experience repository as:

$$\mathscr{E} = \left\{e^{(i)}\right\}_{i=1}^N. \quad (3)$$

## 3.3. Experience-Driven Workflow and Supervised Fine-Tuning

We design a structured web-search workflow in which a base Large Reasoning Model (LRM) $\pi_{\theta_0}$ autonomously executes tasks by interleaving reasoning, tool invocation (SEARCH, SCRAPE), experience retrieval (RETRIEVE), and final answer generation. At execution time, the agent may invoke $\text{RETRIEVE}(q)$ to fetch relevant past experiences from the experience repository $\mathscr{E}$, which are injected as in-context strategic guidance to condition subsequent reasoning and tool-use decisions. This process yields a set of execution trajectories $\mathscr{T} = \{\tau^{(i)}\}_{i=1}^N$, where each trajectory $\tau^{(i)} = (q^{(i)}, h^{(i)}, y^{(i)})$ consists of a user query $q^{(i)}$, the full interaction history $h^{(i)}$ (including retrieved experiences, if any), and the final output $y^{(i)}$. From $\mathscr{T}$, we construct an experience-conditioned SFT dataset by treating each trajectory as a prompt–target pair. The SFT objective is given by:

$$\mathscr{L}_{\text{SFT}} = \mathbb{E}_{(q,h,y)\sim\mathscr{T}}\left[-\log P_\theta(y \mid q, h)\right], \quad (4)$$

which yields an improved policy $\pi_\theta$ that internalizes successful behaviours from the generated experience trajectories.

---

[1] https://huggingface.co/datasets/PersonalAILab/AFM-MHQA-Agent-SFT-Dataset

## 3.4. Evolving Reinforcement Learning

Following SFT, we employ Group Relative Policy Optimization (GRPO), wherein the policy and experience repository $\mathscr{E}$ co-evolve each training iteration refines model parameters while simultaneously updating the strategic knowledge base. Standard GRPO updates policy parameters using group-relative advantages. For a prompt $x$, we sample $K$ rollouts $\mathscr{G} = \{\tau_1, \ldots, \tau_K\}$ and normalize rewards within the group:

$$\hat{A}(\tau) = \frac{r(\tau) - \mu_{\mathscr{G}}}{\sigma_{\mathscr{G}} + \varepsilon} \quad (5)$$

The policy gradient then becomes:

$$\nabla_\theta \mathscr{L}_{\text{GRPO}} = \mathbb{E}_{\tau \sim \mathscr{G}}\left[\nabla_\theta \log \pi_\theta(\tau) \cdot \hat{A}(\tau)\right] \quad (6)$$

However, this *parameter-space optimization* yields zero gradient when all rollouts share the same outcome ($\sigma_{\mathscr{G}} = 0$), eliminating learning signals despite high rollout cost. In web agent settings, where each rollout incurs substantial computational and API cost, discarding these groups leads to significant inefficiency. Therefore, we introduce a complementary *context-space optimization* mechanism that operates by evolving the experience repository $\mathscr{E}$, forming an auxiliary learning channel independent of gradient variance. Since the agent retrieves from $\mathscr{E}$ during inference, an improved repository directly enhances policy behavior through in-context guidance, without modifying $\theta$. Critically, this context-space channel operates on *all* rollouts regardless of reward variance, ensuring no expensive trajectory is wasted.

### 3.4.1. EXPERIENCE EXTRACTION.

At each training step, GRPO generates rollout groups $G_q = \{(\tau_1, r_1), \ldots, (\tau_m, r_m)\}$, where $m$ trajectories attempt the same question $q$. We distill generalizable experience through contrastive analysis across these parallel attempts.

We first compress each trajectory into step-level intended action summaries, distilling the agent's decision rationale at each step:

$$\tau_j^{\text{comp}} = \text{COMPRESS}_\phi(\tau_j) = \{(s_t, a_t^{\text{intent}})\}_{t=1}^{T_j} \quad (7)$$

where $a_t^{\text{intent}}$ captures the high-level goal behind each action. We then construct a contrastive prompt presenting all compressed trajectories with their binary outcome labels $\ell_j \in \{0, 1\}$ indicating failure or success, and extract transferable experience as:

$$E_{\text{new}} = \text{EXTRACT}_\phi\left(q, \{(\tau_j^{\text{comp}}, \ell_j)\}_{j=1}^m\right) \quad (8)$$

In contrast to gradient-based optimization, this experience extraction procedure applies to any reward configuration. All success groups yield distilled winning strategies, all failure groups yield anti patterns, and mixed groups enable direct comparison between effective and ineffective

approaches. As a result, rollout groups with zero reward variance, which are typically discarded in standard parameter space optimization, are transformed into valuable experience contributions.

### 3.4.2. EXPERIENCE CONSOLIDATION.

To control the growth of the experience repository and reduce redundancy, we perform similarity-based consolidation among stored experiences. Let $e^* = \arg\max_{e \in \mathcal{E}} \text{sim}(\mathbf{v}_{e_{\text{new}}}, \mathbf{v}_e)$ denote the most semantically similar existing experience to a newly extracted experience $e_{\text{new}}$, where $\mathbf{v}_e$ represents the embedding of experience $e$. When incorporating $e_{\text{new}}$ into the repository, the update rule is defined as:

$$\mathcal{E} \leftarrow \mathcal{E} \cup \{e_{\text{new}}\} \setminus \{e^* \mid \text{sim}(\mathbf{v}_{e_{\text{new}}}, \mathbf{v}_{e^*}) > \theta\} \quad (9)$$

This consolidation strategy preferentially retains more recent, policy-aligned experiences under high semantic overlap. Because newly extracted experiences capture strategies induced by the current policy state, this replacement mechanism ensures that the experience repository remains compact while continuously evolving its stored knowledge in alignment with the evolving policy throughout training.

## 4. Experiments

### 4.1. Experimental Setup

#### 4.1.1. BENCHMARKS.

We evaluate our method on three representative web-agent benchmarks: **GAIA**[1] (Mialon et al., 2023), **xBench-DeepSearch** (Xbench-Team, 2025), and **HLE**[2] (Phan et al., 2025). These benchmarks collectively cover a wide spectrum of agentic challenges, including multi-hop web search, long-horizon reasoning, and tool invocation.

#### 4.1.2. MODELS AND TRAINING VARIANTS.

We instantiate the policy model with **Qwen-3** (Qwen et al., 2025) at two scales, **4B** and **8B**. For fair comparisons, the experience extractor is held fixed as **Qwen3-30B-A3B** across all experiments, chosen for its sufficient capability to reliably perform structured extraction from rollouts. The policy is embedded in an identical **ReAct**-style agent loop for action selection and tool invocation, so that performance differences across variants are attributable to training dynamics and the experience mechanism rather than to differences in the agent harness or environment.

Our experimental comparison proceeds along two complementary axes. First, to situate EVOLVING ROLLOUTS within the broader landscape of experience-augmented web

---

[1] GAIA: 103 text-only instances (Li et al., 2025c).

[2] HLE: 100 instances randomly sampled from the text-only split.

*Table 1.* Main results on GAIA, xBench-DeepSearch, and HLE (Pass@1). EVOLVING ROLLOUTS outperforms existing experience-augmented web agents (Cai et al., 2025a; Ouyang et al., 2025a; Zhang et al., 2026) on GAIA and xBench under a matched Qwen3-8B backbone, and our 8B model surpasses Qwen3-30B-A3B on average (33.9 vs. 32.2). Baselines marked [†] are reported by Zhang et al. (2026).

| Backbone | Method | GAIA | xBench | HLE | Avg. |
|---|---|---|---|---|---|
| Reference Model | | | | | |
| Qwen3-30B-A3B | | 44.7 | 39.0 | 13.0 | 32.2 |
| Experience-Augmented Web Agents | | | | | |
| | Training-Free GRPO[†] | 29.32 | 26.0 | – | – |
| Qwen3-8B | ReasoningBank[†] | 32.04 | 28.0 | – | – |
| | ExpSeek[†] | 36.89 | 37.20 | – | – |
| EVOLVING ROLLOUTS (Ours) | | | | | |
| Qwen3-4B | RL w/ exp., static | 42.7 | 38.0 | 11.0 | 30.6 |
| | EVOLVING ROLLOUTS | 44.7 | 36.0 | 13.0 | 31.2 |
| Qwen3-8B | RL w/ exp., static | 40.8 | 44.0 | 9.0 | 31.3 |
| | EVOLVING ROLLOUTS | **45.6** | **44.0** | **12.0** | **33.9** |

agents, we benchmark it against representative prior methods under a matched Qwen3-8B backbone. Second, to dissect the contribution of each training stage and isolate the effect of experience context, we evaluate a consistent set of training variants spanning the full post-training pipeline, namely **ReAct** (base instruction-tuned model without post-training), **SFT** on curated agent trajectories, **RL** via **GRPO** (Shao et al., 2024) on top of SFT, and our proposed EVOLVING ROLLOUTS. Additionally, we compare with a much larger model, **Qwen3-30B-A3B**, as a reference to contextualize the performance of our 4B and 8B models.

#### 4.1.3. EVALUATION METRIC.

In the main comparison, we report Pass@1, defined as the fraction of evaluation instances for which the agent successfully completes the task in a single rollout. We adopt Pass@1 in preference to Pass@k because each rollout in our setting incurs substantial cost from two sources: agent trajectories issue numerous calls to commercial web-search APIs, and inference over long, tool-augmented contexts is computationally demanding. Pass@1 further reflects real-world patterns of user interaction, in which the agent is queried once per task rather than re-sampled across multiple trajectories.

### 4.2. Main Results

#### 4.2.1. COMPARISON WITH EXPERIENCE-AUGMENTED WEB AGENTS.

Table 1 presents the main results of EVOLVING ROLLOUTS at both the 4B and 8B model scales, together with representative experience-augmented web-agent baselines and the larger Qwen3-30B-A3B model as a reference for absolute performance. Baseline results for Training-Free GRPO (Cai

*Table 2.* Training-stage decomposition on Qwen3-4B. Adding experience context improves every stage of the pipeline, and the gain grows monotonically with policy strength, peaking when policy and experience are co-evolved (EVOLVING ROLLOUTS). Green deltas in the Avg. column report the absolute improvement over the matching stage without experience context.

| Method | GAIA | xBench | HLE | Avg. |
|---|---|---|---|---|
| Training Stages Without Experience Context | | | | |
| ReAct | 24.3 | 34.0 | 10.0 | 22.8 |
| + SFT | 34.0 | 34.0 | 8.0 | 25.3 |
| + SFT + RL | 41.7 | 35.0 | 10.0 | 28.9 |
| Training Stages With Experience Context | | | | |
| ReAct | 30.1 | 32.0 | 7.0 | 23.0+0.2 |
| + SFT | 35.9 | 32.0 | 11.0 | 26.3+1.0 |
| + SFT + RL (static) | 42.7 | **38.0** | 11.0 | 30.6+1.7 |
| + SFT + RL (EVOLVING ROLLOUTS) | **44.7** | 36.0 | **13.0** | **31.2**+2.3 |

et al., 2025a), ReasoningBank (Ouyang et al., 2025a), and ExpSeek (Zhang et al., 2026) are taken from Zhang et al. (2026), which does not report HLE; the HLE column is therefore left blank for these methods. As shown in Table 1, on GAIA, the 8B EVOLVING ROLLOUTS agent achieves 45.6%, substantially exceeding ExpSeek (36.89%), ReasoningBank (32.04%), and Training-Free GRPO (29.32%). On xBench, it achieves 44.0%, again outperforming all three baselines. Most notably, the 8B variant of EVOLVING ROLLOUTS achieves an average of 33.9%, surpassing the much larger Qwen3-30B-A3B model, which also serves as the fixed experience extractor, at 32.2%. This result shows that the extractor does not constitute an upper bound on downstream policy performance, and further suggests that the gains of EVOLVING ROLLOUTS stem from the co-evolution of policy parameters and experience context, rather than from experience distillation alone. Meanwhile, the 4B variant matches the 30B reference on both GAIA (44.7%) and HLE (13.0%). Within Table 1, holding the experience repository static during RL ("RL w/ exp., static") achieves (31.3 avg at 8B), but jointly evolving the repository with the policy yields an additional 2.6 points (33.9 avg), indicating that the co-evolutionary dynamic, rather than experience conditioning alone.

### 4.2.2. DISSECTING TRAINING STAGES.

To isolate the contribution of each training stage, Table 2 decomposes the Qwen3-4B pipeline into ReAct, SFT, and SFT+RL, with and without our experience mechanism. As shown in Table 2, relative to the SFT+RL baseline without experience (28.9 avg), EVOLVING ROLLOUTS achieves a 2.3-point improvement, reaching 31.2% on average and lifting GAIA from 41.7% to 44.7% (a 3.0-point absolute gain). The gain from SFT to EVOLVING ROLLOUTS exceeds the gain obtained by adding RL alone (static), indicating that experience evolution contributes signal beyond what standard policy-gradient updates recover even when conditioned on a fixed experience repository. A complementary observation

from Table 2 is that the benefit of experience grows monotonically with the strength of the underlying policy: at the ReAct stage, adding experience yields only a marginal gain (22.8 to 23.0); after SFT, the gain widens to a full point (25.3 to 26.3); and after RL, experience contributes the largest improvement, lifting the average from 28.9 to 30.6 with a static repository and to 31.2 with co-evolving experience. This monotonic trend suggests that experience-conditioned guidance becomes increasingly effective as the policy's reasoning and action competencies mature, since a stronger policy is better positioned to interpret and act upon the retrieved strategic cues. We provide a detailed analysis of the mechanisms underlying these gains in Section 5, including controlled ablations on policy–experience co-evolution.

## 5. Ablation Studies

This section investigates the mechanisms underlying EVOLVING ROLLOUTS by disentangling the contributions of policy optimization and experience evolution, and by characterizing how the experience repository evolves throughout training. Our ablations are designed to answer two key questions:(1) whether the performance gains arise from policy learning, experience evolution, or their interaction; and (2) how the composition and utilization of experience change as training progresses.

### 5.1. Decoupling Experience and Policy Contribution

We use two complementary diagnostics to decouple the roles of policy learning and experience evolution. First, Table 3 reports *training-time* ablations using three binary switches: whether training is **with experience** (**w/ Exp**), whether the experiences are updated over training (**Evolve Exp**), and whether the policy parameters are updated (**Evolve Policy**). This isolates the effects of (i) policy-only optimization without experience, (ii) policy optimization with *static experience*, (iii) evolving experience with a frozen policy, and (iv) co-evolving both (EVOLVING ROLLOUTS). Second, Figure 2 provides a *test-time* [2] diagnosis on GAIA by swapping the pairing between checkpoints and experience snapshots (synced vs fixed-policy) without any additional training.

Table 3 shows that evolving only one component is insufficient. Training *with static experience* while evolving the policy improves over the no-experience baseline (30.6 vs 28.9 avg), but co-evolving both achieves the best overall performance (31.2 avg). In contrast, evolving experience while freezing the policy degrades performance sharply (23.0 avg), even below the no-experience baseline.

Figure 2 further decomposes the gains *at test time*. Keeping

---

[2] By *test-time* we mean the fixed-policy pairing (orange line in Figure 2): we keep a fixed policy checkpoint and pair it with experience snapshots from later steps of a completed co-evolution run.

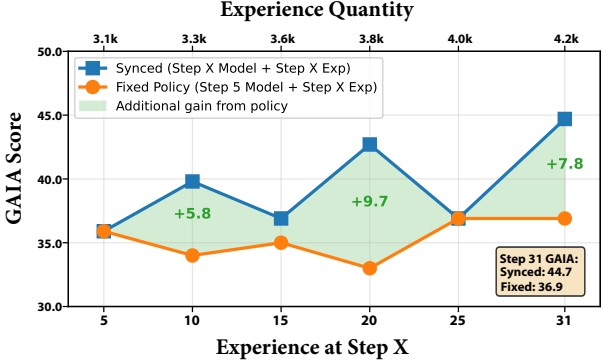

*Figure 2.* Test-time pairing decomposition on GAIA (all checkpoints and experience snapshots are taken from the same co-evolution run). Blue: synced pairing (step-*x* model + step-*x* experience snapshot). Orange: fixed-policy pairing (step-5 model + step-*x* experience snapshot). The green shaded area represents the additional gain from policy evolution under matched pairing.

*Table 3.* Training-time ablations that isolate which components evolve. Rows (top to bottom) correspond to: (1) policy only (no experience), (2) policy + static experience, (3) evolving experience only (frozen policy), and (4) co-evolution (policy + experience).

| w/ Exp | Evolve Exp | Evolve Policy | GAIA | xBench | HLE | Avg. |
|--------|-----------|---------------|------|--------|-----|------|
|        |           | ✓             | 41.7 | 35.0   | 10.0 | 28.9 |
| ✓      |           | ✓             | 42.7 | 38.0   | 11.0 | 30.6 |
| ✓      | ✓         |               | 33.0 | 26.0   | 10.0 | 23.0 |
| ✓      | ✓         | ✓             | **44.7** | **36.0** | **13.0** | **31.2** |

the policy fixed at the step-5 checkpoint while swapping in later experience snapshots (orange) still yields an overall improvement, indicating that the evolved experience itself becomes more useful over training. However, the synced pairing (blue) is consistently higher, and the persistent gap (green shaded region) quantifies the additional gain attributable to policy evolution. Concretely, both curves share the same baseline at step-5 (35.9% GAIA), while the synced pairing exceeds the fixed-policy pairing by +5.8% at step-10 (39.8% vs 34.0%), +9.7% at step-20 (42.7% vs 33.0%), and +7.8% at step-31 (44.7% vs 36.9%).

The frozen-policy training ablation and the fixed-policy test-time pairing reveal a consistent pattern. Decoupling experience evolution from policy evolution, in either direction and at either training or inference time, degrades performance. We attribute this primarily to *policy–experience alignment*. In EVOLVING ROLLOUTS, experience is extracted from rollout groups (e.g., all-fail vs. all-correct) and therefore encodes the transition from failure to success for the policy that produced those rollouts. As the policy improves, the evolving experience increasingly reflects strategies, query formulations, and recovery patterns that are *on-policy* for current checkpoints. The symmetry of this effect across training-time and test-time diagnostics points to a single underlying cause: experience is most beneficial when it stays *on-policy* with the agent model that consumes it.

Co-evolution in EVOLVING ROLLOUTS is precisely the mechanism that maintains this on-policy condition. Every policy update is paired with experience distilled from on-policy rollouts, so the two components co-adapt step by step throughout training. It is this mutual adaptation, rather than experience accumulation *per se*, that EVOLVING ROLLOUTS is designed to exploit, and that the static-experience, and frozen-policy baselines each fail to realize.

### 5.2. Experience Dynamics

Having established the necessity of co-evolution, we next examine how the experience repository itself evolves over training, characterizing its compositional shift, its outcome structure, and the diversity of newly distilled entries.

**Compositional shift toward RL-generated experience.** Over the course of training, the repository grows from 2,996 to 4,197 entries, with a per-step churn rate between 1.0% and 2.1%. Its composition changes substantially during this period: at step 5, SFT-initialized entries account for 93% of the repository (2,942 of 3,172), whereas by step 30 RL-generated entries constitute **31%** (1,274 of 4,162). Figure 3 further shows that these RL-generated entries are not confined to localized clusters but are distributed across the embedding regions also occupied by the SFT initialization, consistent with the interpretation that the policy contributes experiences within the same semantic neighborhoods over which it currently reasons.

**Outcome structure of the repository.** We next examine the repository by outcome type rather than by source. At step 31, it contains 2,162 success strategies (51.5%) and 2,035 failure lessons (48.5%), yielding a near-balanced split without any explicit balancing procedure—a direct consequence of distilling from both all-correct and all-fail rollout groups. Figure 4 indicates that the two outcome types occupy distinct but partially overlapping regions of the embedding space, separating primarily along the first t-SNE axis with a non-trivial transition band between them. This separation is semantically expected, as success strategies and failure lessons describe complementary aspects of the same task distribution; the partial overlap suggests that the distillation procedure recovers both sides of comparable problem instances.

**Diversity of newly distilled experience.** A potential failure mode of coupling RL training to a reused repository is mode collapse, in which the policy converges to a narrow behavioral repertoire and the repository merely reflects this behavior back. We assess this possibility by analyzing the embedding distribution of *newly distilled* entries at each

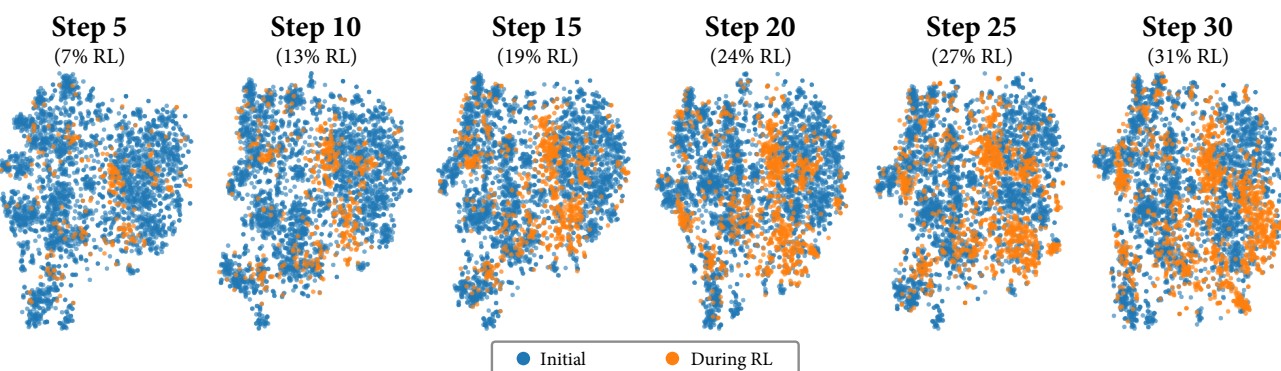

*Figure 3.* t-SNE visualization of experience composition over training (Steps 5 to 30). Blue points represent initialized experiences; orange points represent RL-generated experiences. The percentage of RL experiences grows from 7% to 31% as training progresses, with RL experiences integrating throughout the semantic space rather than forming isolated clusters. This demonstrates the policy's growing contribution to experience through co-evolution.

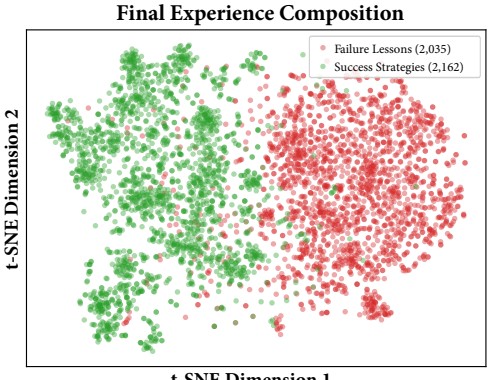

*Figure 4.* Final experience composition by outcome. Green: success strategies. Red: failure lessons. Different outcome occupy distinct but partially overlapping regions of the embedding space.

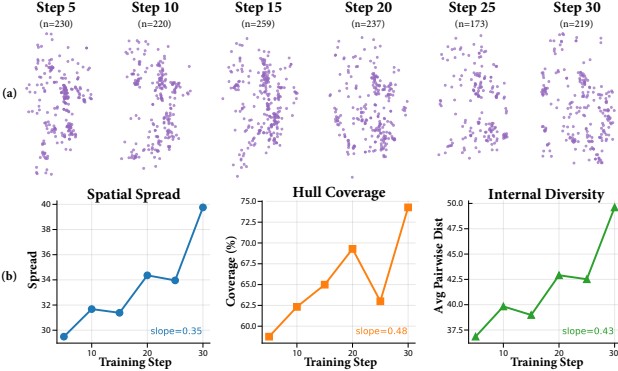

*Figure 5.* (a) t-SNE visualization of newly distilled experiences at each training step, plotted with consistent axis scales across panels. (b) Diversity metrics over training: spatial spread, convex hull coverage, and internal diversity all exhibit upward trends, indicating that the co-evolving policy generates progressively broader experiences rather than collapsing onto a narrow set of recurring patterns.

training step, using three complementary measures: spatial spread (standard deviation), coverage (convex hull area), and internal diversity (mean pairwise distance). All three measures increase monotonically over training (Figure 5): spread rises from 29.5 to 39.8, hull coverage from 59% to 74%, and internal diversity from 36.8 to 49.6. The mean distance from newly distilled entries to the SFT-initialized set increases in parallel from 0.91 to 1.51, indicating that successive snapshots populate regions of the embedding space not covered by the initialization.

Considered jointly, the compositional shift and the monotonic expansion of newly distilled entries indicate that the repository continues to adapt in step with the policy throughout training, rather than stagnating or collapsing onto a fixed distribution. Together with the alignment evidence in §5, these dynamics support the conclusion that on-policy experience co-evolved with the agent policy yields the benefit.

### 5.3. Adaptive Experience Usage and Causal Impact

In EVOLVING ROLLOUTS, experience usage is delegated to the policy: at each step the agent decides whether to invoke

retrieval, and this process converges to a stable usage rate of 4.3% to 5.0% throughout training. The sparsity is emergent, arising from GRPO optimization without any auxiliary objective or explicit budget, and is consistent with the design of EVOLVING ROLLOUTS, whose gains stem from the co-evolution of context and parameter spaces rather than from dense retrieval as in prior experience-based methods. The controlled ablation in Table 3 (rows 2 vs. 4) shows that enabling context-space evolution on top of the parameter-update baseline yields substantial improvements, indicating that sparse but well-timed retrieval is sufficient to drive co-evolution. To assess the causal effect of individual entries, we compare task success rates with and without a given experience on the same tasks. Because retrieval is agentic and is triggered precisely when the policy anticipates difficulty, the activated tasks are systematically harder: their mean baseline success rate is 37.3%, well below the population

average. Under this hard-instance conditioning, 24.1% of 696 analyzed experiences exhibit positive causal impact, with the top entries achieving success-rate improvements exceeding +0.67 and reaching 100% success on tasks with baseline rates near 32%. At the system level, rows 1 vs. 2 of Table 3 confirm that the availability of retrieval consistently improves overall performance, so per-experience deltas and benchmark-level gains capture complementary facets of the same mechanism. Qualitative inspection of the highest-impact entries shows that they encode reusable problem-solving patterns rather than task-specific heuristics, capturing explicit verification strategies (e.g., cross-referential reasoning chains) and systematic remedies for recurring failure modes (e.g., temporal ambiguity in historical milestones). Together with the emergent sparse usage pattern, these results indicate that the most effective experiences function as transferable reasoning primitives, deployed precisely where targeted guidance is most valuable.

## 6. Conclusion

We presented EVOLVING ROLLOUTS, a framework that co-evolves policy parameters and an experience repository during RL training. By distilling reusable strategies from zero-variance rollout groups (all-success or all-failure), which yield no gradient signal in standard GRPO, EVOLVING ROLLOUTS recovers learning signal that would otherwise be discarded. Experiments show that EVOLVING ROLLOUTS enables Qwen3-8B to surpass Qwen3-30B-A3B in average performance across GAIA, xBench, and HLE, and enables Qwen3-4B to match Qwen3-30B-A3B on GAIA and HLE, while ablations confirm that co-evolution is essential for performance improvement.

## Impact Statement

This paper presents a reinforcement learning framework for training web-based agents with improved sample efficiency. Our work aims to advance the field of Machine Learning, particularly in the area of agentic systems that interact with real-world web environments.

We acknowledge that more capable web agents could potentially be misused for tasks such as automated misinformation gathering or unauthorized data collection. However, these concerns are not unique to our method and apply broadly to web agent research. Our framework does not introduce new capabilities beyond what existing web agents can achieve; rather, it improves the training efficiency of such systems. We believe the benefits of more sample-efficient training, which reduces computational costs and environmental impact, outweigh these general concerns.

We release our training framework to support reproducibility and encourage responsible development in this area.

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

# A. Experiment Setup

## A.1. Experience Extraction Details

The quality of extracted experiences is critical for effective context-space optimization. We design the extraction process to maximize transferability and minimize redundancy.

**Trajectory Compression.** Raw web-agent trajectories are dominated by verbose observations (e.g., full webpage HTML) and redundant intermediate reasoning (Lu et al., 2026), which are unsuitable for direct reuse as transferable experience. We compress each trajectory independently using a frozen LLM (Qwen3-30B-A3B (Yang et al., 2025a)), which outputs a structured JSON summary capturing the essential decision logic.

The compression schema extracts: (1) an *overall plan* summarizing the global strategy, (2) a sequence of *steps*, each containing the high-level intent, tool usage with query formulation and URL selection rationale, and a condensed observation summary, and (3) *key takeaways* encoding actionable strategy insights.

## A.2. Training Data

We construct the training data for both supervised fine-tuning (SFT) and reinforcement learning (RL) by sampling questions from a diverse collection of web-agent benchmarks, including WebWalkerQA, MHQA, TaskCraft, ASearcher, WebShaper, and 2WikiMultiHopQA. From this combined pool, we randomly sample 1,000 questions for the SFT stage and a separate set of 1,000 questions for the RL stage, ensuring no overlap between the two sets. This separation prevents data leakage and ensures that the RL phase encounters novel questions that test generalization beyond the SFT distribution. The diversity of source datasets exposes the agent to a wide range of web interaction patterns, from simple fact retrieval to complex multi-hop reasoning chains.

# B. Case Study: Evolving Experience

---

**Trajectory for Experience Retrieval**

**Pre-retrieval State Summary:**

- Goal: Extract command from last video in 2018 Replit VSCode blog post.

- Issue: Searched blog post (`https://blog.replit.com/intel`) is text-only—no video or command found.

- Hypothesis: URL may be malformed or truncated in search results.

- Blocked: Cannot verify content without correct URL.

**Tool Call:**

```
{
  "name": "retrieve_experience",
  "arguments": {
    "query": "How to verify the
    correct URL structure for
    Replit blog posts when search
    results show truncated
    or incorrect URLs?"
  }
}
```

**Retrieved Experience (Strategy Guidance):**

- **Success Strategy 1**: Direct homepage validation over secondary sources—crawl official domain to confirm content.

- **Success Strategy 2**: Standard URL pattern identification (e.g., `/post/title`) reduces ambiguity.

- **Failure Lesson 3**: Avoid broad queries without domain validation.

- **Failure Lesson 4**: Never trust URL patterns alone—always validate page content.

---

---

**Evolving Experience**

i). **Retrieved memory (Top-1 most relevant):**
Filter for Explicit Country Mentions Over Incidental References

ii). **New memory title:**
Historical Context Integration for Geopolitical Nuance

iii). **New memory content:**
Incorporating historical country labels (e.g., "West German") into verification prevents anachronistic misclassification.

---

## B.1. Trajectory Compression Prompt

**Trajectory Summarizer Prompt**

*You are an AI Trajectory Summarizer. Compress verbose web search agent logs into strategic JSON summaries.*

**TASK:** Convert detailed trajectory to compact concise step-wise JSON capturing decision logic, eliminating redundancy.

**OUTPUT JSON structure:**

```
{
  "overall_view": "1-2 sentence global plan",
  "steps": [{
    "step": 1,
    "high_level_intent": "strategic goal",
    "action": {
      "tool": "tool_name_if_applicable",
      "query_strategy":
        "search formulation logic",
      "url_selection_strategy":
        "URL prioritization logic",
      "extraction_focus":
        "target information"
    },
    "observation_summary":
      "one-sentence key findings",
    "key_information_extracted":
      ["core fact"]
  }, ...],
  "key_takeaways": ["strategy insights"]
}
```

**RULES:**
1. Transform queries to search logic descriptions, not raw text
2. Convert observations to synthesized information states
3. Preserve goal decomposition and tool selection rationale
4. Omit system prompts, HTML, boilerplate, repetitive thinking
5. Identify successful heuristics and efficiency patterns
6. Ensure takeaways are actionable and transferable

**EXAMPLE TRANSFORMATIONS:**
• Query "MIT Review 2025 AI trends" →
   "Combined publication, timeframe, topic for precise targeting"
• Long observation text →
   "Confirmed partnership focuses on counter-drone technology"

*Now compress this trajectory.*

---

The compression rules emphasize transforming raw queries into search logic descriptions, converting verbose observations into synthesized information states, preserving goal decomposition and tool selection rationale, and omitting HTML content and repetitive reasoning. This reduces context length by 5–10× while preserving strategic content. Compression is parallelized across all trajectories in the rollout group via vLLM (Kwon et al., 2023) batched inference, minimizing wall-clock overhead.

**Group Aggregation and Contrastive Prompt Design.**    During the RL stage, experience extraction is performed over groups of reward-labeled trajectories. These outcome labels are incorporated into contrastive prompts that explicitly contrast successful and failed attempts, or alternatively summarize common correct behaviors and recurring incorrect patterns that should be avoided.

The contrastive extraction prompt aggregates all compressed trajectories from a rollout group with their outcome labels. For failed attempts, the expected answer is included when available.

---

**Group Extraction Prompt Template**

```
You have multiple attempts to solve this problem, which may include both successful
and failed executions.  Extract 1-5 experiences that contain generalizable
strategies and lessons learned from these attempts.

PROBLEM: {question}

ATTEMPT1 (SUCCESS): {compressed_trajectory_1}
ATTEMPT2 (FAILURE): {compressed_trajectory_2}
EXPECTED: {expected_answer}
...

Extract experiences in this format:

EXPERIENCE 1:
TITLE: <concise strategy name or lesson>
DESCRIPTION: <one sentence summary>
CONTENT: <detailed transferable strategy or lesson>

Focus on:
- WHY successful approaches worked
- WHAT went wrong in failed attempts
- Generalizable patterns across attempts
```

---

The group extraction prompt naturally handles all outcome configurations. For all-success groups, the LLM identifies common winning strategies; for all-failure groups, it extracts anti-patterns and lessons from mistakes. This ensures every rollout group contributes to the experience repository, regardless of its utility for gradient-based policy optimization.

This ensures every rollout group contributes to the experience repository, regardless of its utility for gradient-based optimization.

**Experience Embedding and Dual Representations.**    After extraction, each experience is embedded to support efficient semantic retrieval. Specifically, we compute two complementary dense representations per experience using a lightweight encoder (Qwen3-0.6B-Embedding (Yang et al., 2025a)): an *experience embedding* $\mathbf{v}_e \in \mathbb{R}^d$, encoded from the concatenation of the experience title and description to capture high-level strategic semantics for query-to-experience matching, and a *source problem embedding* $\mathbf{v}_e^{\mathrm{src}} \in \mathbb{R}^d$, encoded from the original problem text to model problem-type similarity between the current task and the experience's originating context. This dual-embedding design underpins our retrieval mechanism: given an agent query representation $\mathbf{v}_q$ and a task problem representation $\mathbf{v}_p$, experiences are ranked by jointly aggregating query-to-experience similarity $\mathrm{sim}(\mathbf{v}_q, \mathbf{v}_e)$ and problem-to-source similarity $\mathrm{sim}(\mathbf{v}_p, \mathbf{v}_e^{\mathrm{src}})$, ensuring that retrieved experiences are both semantically aligned with the agent's current reasoning state and contextually relevant to the target problem type.

Each extracted experience is formalized as a structured knowledge unit with a well-defined schema, designed to support reliable retrieval and effective in-context utilization. Specifically, each experience $e \in \mathscr{E}$ comprises the following fields:

Each extracted experience is formalized as a structured knowledge unit comprising both semantic content and dense vector representations for efficient retrieval.

**Semantic Content.**    The textual component of each experience $e \in \mathscr{E}$ consists of:

- **Title**: A concise strategy identifier that abstracts the core decision pattern (e.g., "Cross-Reference Official Sources for Temporal Verification").
- **Description**: A declarative summary specifying applicability conditions and contextual triggers for the strategy.
- **Content**: A detailed reasoning template encoding transferable procedural logic and concrete guidance for in-context application.
- **Source Problem**: The original question from which the experience was extracted, preserving provenance for problem-type matching.
- **Outcome Label**: A binary indicator (`success` $\in \{0, 1\}$) denoting whether the experience derives from a successful or failed trajectory, used to distinguish winning strategies from cautionary lessons during retrieval formatting.

**Retrieval Trigger.**   The agent autonomously decides when to retrieve based on its reasoning state. Typical triggers include:

- Encountering an unfamiliar problem type
- Reaching a reasoning impasse after failed search attempts
- Seeking verification strategies before committing to an answer

**Experience Formatting.**   Retrieved experiences are formatted as structured guidance:

> — *Retrieved Experiences from Memory Bank* —
>
> *[Experience 1]* **Title**: *Cross-Reference Official Sources*
> **Strategy**: *When verifying institutional information, prioritize official .edu or .gov domains over third-party aggregators...*
>
> *[Experience 2] ...*
>
> — *End of Retrieved Experiences* —
>
> *Based on these strategies, refine your approach for the current task. Note: Experiences provide HOW to reason, not factual answers—verify all claims through web search.*

### B.2. Training Pipeline

The complete training pipeline consists of three phases:

**Phase 1: Cold-Start Repository Construction.**   We process $N$ trajectories from existing web-agent datasets to construct $\mathscr{E}_0$. Each trajectory is passed through the extraction pipeline independently (without contrastive grouping, as trajectories come from different questions). This phase is executed once before training.

**Phase 2: Supervised Fine-Tuning.**   Using the base LRM $\pi_{\theta_0}$ with access to $\mathscr{E}_0$, we generate SFT trajectories on a training question set. The model learns to:

- Execute the structured workflow (reason $\rightarrow$ search $\rightarrow$ scrape $\rightarrow$ answer)
- Invoke RETRIEVE at appropriate reasoning states
- Interpret and apply retrieved experiences to the current task

The SFT phase produces $\pi_\theta$ with basic competency in experience-augmented reasoning.

**Phase 3: Reinforcement Learning with Evolution.**   We apply GRPO on held-out questions while dynamically evolving $\mathscr{E}$. At each step:

1. Generate $m$ rollouts per question with retrieval from current $\mathscr{E}$
2. Compute rewards via LLM-as-a-Judge
3. **Context-space**: Extract experiences from all groups, consolidate, and forget
4. **Parameter-space**: Update $\pi_\theta$ via GRPO on non-zero-variance groups
5. Publish updated $\mathscr{E}$ for next iteration

**Implementation Details.**   We implement the supervised fine-tuning (SFT) stage using the `ms-swift` (Zhao et al., 2024) framework, which provides scalable support for instruction tuning and experience-augmented data pipelines on large reasoning models. The reinforcement learning stage is implemented with `veRL` (Sheng et al., 2024), a distributed RL framework that supports group-based policy optimization (GRPO) with efficient rollout generation and reward aggregation. All experiments are conducted on NVIDIA H20 GPUs.

### B.3. Computational Considerations

**Extraction Overhead.**   Experience extraction adds computational cost, but this is amortized across future rollouts that benefit from the enriched repository. We extract experiences asynchronously with policy updates to minimize wall-clock overhead.

**Repository Size Management.** The merge threshold $\theta$ controls repository growth. With $\theta = 0.9$, we observe approximately linear growth early in training (diverse experiences) followed by sublinear growth as the repository saturates with high-quality strategies and new experiences increasingly merge with existing ones.

### B.4. Hyperparameters

Table 4 summarizes key hyperparameters for the evolution mechanisms.

*Table 4.* Hyperparameters for experience evolution.

| Parameter | Symbol | Value |
|---|---|---|
| Merge threshold | $\theta$ | 0.90 |
| Retrieval top-$k$ | $k$ | 5 |
| Dual similarity weight | $\alpha$ | 0.5 |
| Success threshold | - | 0.5 |
| Experiences per extraction | - | 1–5 |
| Rollouts per question (GRPO) | $m$ | 8 |

**Sensitivity Analysis.** The merge threshold $\theta$ presents a trade-off: higher values preserve more distinct experiences but risk redundancy; lower values aggressively consolidate but may lose nuance. We find $\theta \in [0.85, 0.95]$ works well across benchmarks. The forgetting mechanism is intentionally conservative (requiring unanimous failure) to prevent premature deletion of potentially useful strategies.

## C. Software Frameworks and Licenses

*Table 5.* Software components and licenses.

| Component | Role | License |
|---|---|---|
| `ms-swift` (Zhao et al., 2024) | SFT | Apache License 2.0 |
| `veRL` (Sheng et al., 2024) | RL | Apache License 2.0 |
| `Qwen-3` (Yang et al., 2025a) | Backbone | Apache License 2.0 |

This section summarizes the major frameworks used in our training pipeline and their corresponding licenses. All components are selected to ensure reproducibility and compliance with open-source research practices.

