# OpenReview forum: "EVOLVING ROLLOUTS: Harnessing Historical Experience for Web Agent Evolution in Reinforcement Learning"
_ICML.cc/2026/Conference — ICML 2026 regular_

### Official Review · Reviewer_K6d7 · 2026-02-15

**Soundness:** 3
**Presentation:** 2
**Significance:** 3
**Originality:** 3
**Overall Recommendation:** 4
**Confidence:** 3

**Summary:**

The paper proposes "Evolving Rollouts," a framework designed to improve the sample efficiency of Agentic Reinforcement Learning, specifically within the context of web search agents trained via Group Relative Policy Optimization (GRPO). The authors identify a critical inefficiency in standard GRPO: when a group of rollouts yields zero reward variance (i.e., all fail or all succeed), the gradient signal vanishes, causing the model to discard potentially valuable interaction data. To address this, the authors introduce a co-evolutionary framework where an "experience repository" is updated alongside the policy. Instead of discarding zero-variance groups, the method uses a frozen LLM to distill these trajectories into textual "experiences" (strategies or failure lessons). These experiences are then retrieved and used as in-context guidance for future rollouts. Empirical results on GAIA, XBench, and HLE suggest that a 4B parameter model trained with this method can match the performance of a 30B baseline.

**Compliance With Llm Reviewing Policy:**

Affirmed.

**Final Justification:**

Thank you for the thorough and well-structured rebuttal. The additional experiments and clarifications have addressed my concerns satisfactorily, particularly regarding computational overhead and the distinction from distillation-based approaches. I am happy to raise my score accordingly.

**Key Questions For Authors:**

## **Questions**
**Q1. Relation to Knowledge Distillation (KD)**: The proposed framework structurally resembles a form of "Contextual" or "In-context" Knowledge Distillation, where a 30B teacher provides strategic insights to a 4B student. Could the authors clarify how this approach fundamentally differs from, or outperforms, traditional offline distillation (e.g., SFT on teacher-generated trajectories)? An empirical comparison with standard KD baselines would significantly clarify the unique value of the evolutionary repository. A response showing that the co-evolutionary loop outperforms static distillation would increase my score for originality and significance.

**Q2. Generalizability of Fixed Parameters**: Were the hyperparameters in Table 3 held constant across all benchmarks (GAIA, XBench, HLE), or were they tuned per task? If they were constant, it speaks to the robustness of the method; if not, it suggests a potential sensitivity that should be discussed. Clarification on this point will help evaluate the generalizability of the proposed framework.

**Q3. Empirical Evidence for Hyperparameters and Repository Scale**: While Table 3 lists key hyperparameters, the paper lacks detailed ablation studies for critical variables such as the Merge threshold ($\theta$) and Retrieval top-k ($k$). Furthermore, could the authors provide the exact number of experiences in the repository at the end of training for each benchmark? Detailed sensitivity curves and repository size statistics would strengthen the soundness of the methodology and address concerns regarding reproducibility.

**Q4. Computational Efficiency and Overhead**: The framework involves high-frequency inference from a 30B model for extraction and additional RAG-based retrieval steps. What is the total computational overhead (e.g., wall-clock time or TFLOPS) compared to a vanilla GRPO pipeline? If the authors can demonstrate that the sample efficiency gains significantly outweigh the increased per-step computational cost, it would greatly enhance my assessment of the paper's practical significance.

**Limitations:**

While the authors have addressed several technical aspects of their framework, the discussion on limitations could be strengthened by explicitly tackling the following points:

**L1. Computational Cost of Experience Extraction**: The primary advantage of the proposed method is sample efficiency in terms of environment interactions. However, this comes at the cost of high-frequency inference from a significantly larger teacher model (30B) during the training loop. The authors should discuss the net resource trade-off; specifically, whether the total computational budget (GPU hours/TFLOPS) remains advantageous when the overhead of the 30B extractor is factored in alongside the 4B model's training.

**L2. Teacher-Induced Performance Ceiling**: The framework relies heavily on the 30B model to diagnose failures and extract strategies. This creates an inherent "ceiling effect," where the 4B agent's strategic reasoning is bounded by the capabilities of the larger model. The authors should address whether the agent can eventually transcend the teacher’s logic or if it remains permanently dependent on the quality of the teacher's distillation.

**Strengths And Weaknesses:**

## **Strengths**
**S1. Originality in Addressing Vanishing Gradients**: The paper provides a novel solution to the zero-reward-variance problem in GRPO. By converting discarded rollout groups into "strategic experiences" through textual distillation, the framework successfully recovers learning signals from samples that would otherwise be wasted.

**S2. Enhanced Sample Efficiency**: The method demonstrates a sophisticated use of data by moving beyond episodic training. By bridging parameter-space and context-space optimization, the agent achieves higher performance with fewer environment steps, which is critical for expensive web-interaction tasks.

**S3. Empirical Significance**: The results show that a 4B parameter model, when augmented with the evolving experience repository, can match or exceed the performance of a 30B baseline. This is highly significant for the development of deployable, smaller-scale agents that maintain high-level reasoning capabilities.


## **Weaknesses**
**W1. Computational Overhead of the Extractor**: A major concern regarding soundness is the reliance on a large 30B teacher model for real-time experience extraction during the RL loop. The authors should provide a detailed comparison of the total computational cost (e.g., TFLOPS or GPU hours) relative to a naive training pipeline to justify if the sample efficiency gains outweigh the extraction overhead.

**W2. Performance Bottlenecks (Ceiling Effect)**: Since the 4B agent’s strategies are distilled from the 30B model, the agent’s potential performance appears to be inherently capped by the teacher's capabilities. There is a lack of discussion on how the agent might evolve beyond the teacher's reasoning limits.

**W3. Insufficient Comparative Analysis**: The paper lacks direct empirical comparisons with concurrent "experience-driven" agentic models mentioned in the related work. Furthermore, since this method functions as a form of contextual knowledge distillation, it is necessary to compare it against standard distillation baselines (e.g., SFT on teacher trajectories) to demonstrate the unique value of the evolutionary repository.

**W4. Reproducibility and Parameter Sensitivity**: While Appendix B.4 lists key hyperparameters, the lack of rigorous ablation studies or sensitivity curves (e.g., varying the merge threshold $\theta$ or retrieval $k$) makes it difficult to assess the robustness of the system. Additionally, the paper does not explicitly state whether the same hyperparameter configuration was used across all benchmarks, which raises concerns about the generalizability of the results.

---

> ### Author Rebuttal · Authors · 2026-03-31
>
> We greatly appreciate the reviewer for recognizing our originality and meaningful empirical contributions.
>
> ### **1. Computation Overhead (W1, Q4)**
>
> The extractor overhead is negligible, and it fundamentally recovers otherwise wasted signals from policy RL compute. We estimate FLOPs using standard approximations: decoding costs $\approx 2N$ per token, and RL training (forward + backward) costs $\approx 6N$ per token, giving **$8N$ per token total** for the base pipeline.
>
> * **Base RL Cost:** The policy generates 7,680 sequences (8 rollouts × 32 batch × 30 iterations) at an average length of $3.3 \times 10^5$, totaling $2.53 \times 10^9$ tokens. This yields $8.11 \times 10^{19}$ FLOPs for Qwen3-4B and $1.62 \times 10^{20}$ FLOPs for Qwen3-8B.
> * **Experience Extraction Cost:** The Qwen3-30B-A3B teacher runs forward-pass inference only ($2N$ per token). It is invoked *only* on zero-variance reward groups (~30% of rollout groups)—these are trajectories that contribute zero gradient in standard GRPO and represent purely wasted compute. With ~3B active MoE parameters and $7.60 \times 10^8$ effective tokens, the cost is $4.56 \times 10^{18}$ FLOPs.
>
> **Conclusion on Compute:** As shown in the table below, this represents just a **~5.6% overhead for Qwen3-4B** and **~2.8% for Qwen3-8B**. Critically, without Evolve Rollouts, ~30% of all rollout compute is entirely wasted (yielding zero learning signal). The extractor reclaims this dead compute at a fractional cost, and its effectiveness is validated across different models over three benchmarks, confirming the recovered signal is general and impactful.
>
> **Table 1: Computation Overhead Comparison across Models**
>
> | Model | FLOPs: Base RL ($8N$/token) | FLOPs: Experience Extraction ($2N$/token) | Additional Compute |
> | :--- | :--- | :--- | :--- |
> | Qwen3-4B | $8.11 \times 10^{19}$ | $4.56 \times 10^{18}$ | **+5.6%** |
> | Qwen3-8B | $1.62 \times 10^{20}$ | $4.56 \times 10^{18}$ | **+2.8%** |
>
> ---
>
> ### **2. Differences from Distillation (W2, W3, Q1)**
>
> Regarding the reviewer’s concern about the distillation ceiling effect: **Evolving Rollouts is not simple distillation.** It extracts learning signals from wasted long-horizon trajectories into the context space, enabling *co-evolution* between the context and parameter spaces during training.
>
> * **Breaking the Ceiling Effect:** Our Qwen3-8B implementation reaches **33.9%**, outperforming its Qwen3-30B-A3B experience extractor teacher (**33.2%**). This confirms that no ceiling is imposed by the extractor and further validates that our mechanism relies on co-evolution (as shown in Figure 2) rather than mere distillation.
> * **Comparison to a Pure Distillation Baseline:** The SFT trajectory is distilled directly from Qwen3-30B-A3B; hence, the SFT entry in Table 1 serves as a pure distillation baseline. It attains only **26.3%** averaged performance across GAIA, xBench, and HLE, which is substantially lower than the **31.2%** achieved by Evolving Rollouts on the same benchmarks. This verifies the efficacy of our co-evolution design over basic distillation.
> * **Context within Experience-Driven Methods:** When we developed Evolving Rollouts, most existing approaches focused on benchmarks like LoCoMo and AFLWorld, whose agent settings differ substantially from a web search environment. We acknowledge the concurrent work ExpSeek [1], which adopts an experience-driven paradigm tailored specifically for web search agents, and benchmarks relevant methods on web search tasks including GAIA and xBench.
>
> ---
>
> ### **3. Hyperparameters & Sensitivity (W4, Q2, Q3)**
>
> We thank the reviewer for raising the question about hyperparameter settings. **All hyperparameters are consistent** throughout all training stages (SFT and RL) and across all tests on all benchmarks (GAIA, xBench, and HLE). We will explicitly mention this in the appendix.
>
> To test the robustness of Evolving Rollouts, we conducted sensitivity experiments on the merge threshold and retrieval top-$k$ using the 4B model:
>
> * **Merge Threshold Sensitivity:** When reducing the threshold from 0.9 to 0.6, the final experience repository size reduced from 4,006 to 3,472. However, overall performance remained robust with only a **0.1% average drop**, which falls within the margin of expected noise.
> * **Retrieval Top-$k$ Sensitivity:** We evaluated different retrieval top-$k$ values:
>     * Setting $k = 3$ improved average performance by **0.2%** over the original configuration ($k = 5$).
>     * Increasing $k$ to 10 caused a **0.7% performance decline**. This trend is expected: concatenating ten experience entries introduces an excessively long context, which distracts the model and shifts its focus away from the target task.

---

> > ### Author Rebuttal · Reviewer_K6d7 · 2026-04-02
> >
> > Thank you for the thorough and well-structured rebuttal. The additional experiments and clarifications have addressed my concerns satisfactorily, particularly regarding computational overhead and the distinction from distillation-based approaches. I am happy to raise my score accordingly.

---

> > > ### Author Response · Authors · 2026-04-02
> > >
> > > Thank you for your positive feedback. Your constructive comments have helped us improve our work.

---

### Official Review · Reviewer_eN7H · 2026-03-05

**Soundness:** 2
**Presentation:** 3
**Significance:** 3
**Originality:** 3
**Overall Recommendation:** 4
**Confidence:** 3

**Summary:**

In agent reinforcement learning, most traditional works rely on outcome based binary rewards, leading to the problem of sparse rewards during model training. To address this issue, the authors use an RL method called EVOLVING ROLLOUTS to alleviate the vanishing policy gradient problem within a batch. Specifically, it extracts reusable signals from trajectories that have zero reward variance. This method jointly optimizes the model policy alongside a concurrently evolving experience bank, and the authors validate the effectiveness of the method in downstream tasks.

**Compliance With Llm Reviewing Policy:**

Affirmed.

**Final Justification:**

Thanks to the authors for their response. The rebuttal has addressed my concerns, so I have raised my score.

**Key Questions For Authors:**

(1) Similar to the weaknesses mentioned above, if a model of the same scale as the agent (such as 4B) is used to replace the 30B model for experience compression and extraction, by how much would the overall performance and experience quality decrease?

(2) Given the continuously expanding experience bank during training, at what point will system retrieval latency become a performance bottleneck during the inference process?

**Limitations:**

yes

**Strengths And Weaknesses:**

Strengths

(1) This method effectively utilizes expensive interaction trajectories that are typically discarded in traditional GRPO due to zero reward variance.

(2) It enables dual optimization in both the parameter space and the prompt space.

(3) To control the growth of the experience bank, the authors implement similarity based integration, thereby reducing redundancy.

Weaknesses

(1) The baselines are too weak, lacking relevant baselines based on RL methods, such as GiGPO [1] and SPEAR [2].

(2) The Trajectory Compression process relies on a more powerful model, Qwen 3 30B A3B. It remains unclear whether the evolution process can still be achieved if it relies solely on the policy model itself instead of a stronger teacher model.

(3) The experiments mainly focus on the 4B version of the Qwen 3 model family, lacking generalization validation across more scales or architectures.

[1] Feng L, Xue Z, Liu T, et al. Group-in-group policy optimization for llm agent training[J]. arXiv preprint arXiv:2505.10978, 2025.

[2] Qin Y, Tan X, He Z, et al. Learn the ropes, then trust the wins: self-imitation with progressive exploration for agentic reinforcement learning[J]. arXiv preprint arXiv:2509.22601, 2025.

---

> ### Author Rebuttal · Authors · 2026-03-31
>
> We thank the reviewers for recognizing our effective integration of active experience retrieval into GRPO training, along with consistent benchmark improvements and comprehensive ablations.
>
> - **baselines and generalizability** (W1, Q3)
>
> To address the concern about baselines and generalizability, we conducted additional experiments. First, we added comparisons with recent experience-augmented web agent methods, including Training-Free GRPO, ReasoningBank, and ExpSeek, under both Qwen3-32B and Qwen3-8B base models. Specifically, we include results reported in ExpSeek (Wu et al., 2025). Second, we extended our Evolving Rollouts pipeline to Qwen3-8B to validate generalization across model scales. All results are summarized in the updated table below.
>
> For Qwen3-8B, the model without evolve (SFT + RL) obtains 40.8%, 33.0%, and 6.0% on GAIA, xBench, and HLE, respectively (Avg. 26.6%). Incorporating static experience raises performance to 40.8%, 44.0%, and 9.0% (Avg. 31.3%), while the full Evolving Rollouts pipeline further improves results to 45.6%, 44.0%, and 12.0% (Avg. 33.9%), surpassing the significantly larger Qwen3-30B-A3B model (Avg. 32.2%).
>
> Compared with experience-augmented methods, our Qwen3-8B with Evolving Rollouts substantially outperforms ExpSeek on a comparable Qwen3-8B base (45.6% vs. 36.89% on GAIA; 44.0% vs. 37.20% on xBench), and even exceeds ExpSeek on the much larger Qwen3-32B base (45.6% vs. 43.88% on GAIA; 44.0% vs. 42.00% on xBench).
>
> Consistent with our earlier results on Qwen3-4B, these outcomes confirm that the effectiveness of Evolving Rollouts generalizes reliably across different model scales, and that it compares favorably against recent experience-augmented agent methods.
>
>
> | Method | GAIA | xBench | HLE | Avg. |
> | :--- | :---: | :---: | :---: | :---: |
> | *Reference Model* | | | | |
> | Qwen3-30B-A3B | 44.7 | 39.0 | 13.0 | 32.2 |
> | **Web Agent with Experience Tool (Qwen3-32B base)** | | | | |
> | Training-Free GRPO | 36.89 | 28.20 | -- | -- |
> | ReasoningBank | 33.01 | 36.33 | -- | -- |
> | ExpSeek | 43.88 | 42.00 | -- | -- |
> | **Web Agent with Experience Tool (Qwen3-8B base)** | | | | |
> | Training-Free GRPO | 29.32 | 26.00 | -- | -- |
> | ReasoningBank | 32.04 | 28.00 | -- | -- |
> | ExpSeek | 36.89 | 37.20 | -- | -- |
> | **Evolving Rollouts (Ours, Qwen3-4B)** | | | | |
> | + SFT + RL (w_exp but static) | 42.7 | 38.0 | 11.0 | 30.6 |
> | + SFT + RL (evolving) | **44.7** | 36.0 | **13.0** | **31.2** |
> | **Evolving Rollouts (Ours, Qwen3-8B)** | | | | |
> | + SFT + RL (w_exp but static) | 40.8 | 44.0 | 9.0 | 31.3 |
> | + SFT + RL (evolving) | **45.6** | **44.0** | **12.0** | **33.9** |
>
>
> - **Experience Model** (W2, Q1):
>   - We appreciate this insightful question regarding replacing the strong 30B teacher model with the policy model itself for experience extractor. Since the idea of Evolving Rollouts is to recycle the otherwise wasted rollouts into learning signals to context space and forming co-evolution between context space and parameter space. We fixed the 30B-A3B extractor across all experiments as a stable, competent tool to eliminate external noise and ensure fair controlled validation of our core idea of context and parameter space co-evolving, so we did not explore on the extractor settings in this work.
>
>
>    - But regarding to reviewer's proposal, our hypothesis is the experience model can be replaced or at least there exist a sweet spot related to current model perofrmance during evolution. During rebuttal we conducted thorogh experiments on Qwen3-8B model and found it's already outperforms the 30B extractor's performance, so we strongly believe at least there already exist a sweet spot for the extractor-policy pairing during Evolving Rollouts training.
>
> - **Q2**: We understand reviewer's concern about latency when our method scales. In our framework, retrieval uses cosine similarity computed via matrix dot product. The computation is constant-time O(1) and independent of the experience bank size. Therefore, the growing experience bank will not introduce additional latency or become a bottleneck during inference.

---

> > ### Author Rebuttal · Reviewer_eN7H · 2026-04-02
> >
> > Thanks to the authors for their response. The rebuttal has addressed my concerns, so I have raised my score.

---

> > > ### Author Response · Authors · 2026-04-02
> > >
> > > Thank you for your positive feedback. Your constructive comments have helped us improve our work.

---

### Official Review · Reviewer_eAMU · 2026-03-14

**Soundness:** 3
**Presentation:** 2
**Significance:** 3
**Originality:** 2
**Overall Recommendation:** 4
**Confidence:** 3

**Summary:**

This paper proposes Evolving rollouts, which proposes to incorporate trajectory retrieval into GRPO training process using an experience curation mechanism. By using retrieved experiences as in-context guidance, the method better leverages zero reward variance rollouts, leading to better performance and an additional axis for performance scaling (the evolving memory bank).

**Compliance With Llm Reviewing Policy:**

Affirmed.

**Final Justification:**

While the components such as evolving memory bank and GRPO are standard components, the authors successfully incorporate retrieving from an actively maintained experience bank into GRPO training as an effective co-evolving setup. My initial concerns were with the experiments being restricted to one model, Qwen-3 4B, which the authors have addressed with additional results showing positive results on the larger 8B model. And while it seems that retrieval is triggered rarely, this helps to supply critical context to the model in a co-evolving way as shown by the empirical results, which is an interesting finding in itself.

**Key Questions For Authors:**

- From what I can understand, the experience retrieval is selectively done by the policy. How is this done? Does the behavior simply arise as a result of GRPO training? Did the authors try enforcing retrieval to be always on?
- Did the authors try different designs for the input to the embedding model?
- Is the cold-start dataset the same as the SFT dataset?

**Limitations:**

yes

**Strengths And Weaknesses:**

Strengths
- While the components such as evolving memory bank and GRPO are standard components, the authors successfully incorporate retrieving from an actively maintained experience bank into GRPO training as an effective co-evolving setup.
- Performance gains are consistent across 3 benchmarks, and comprehensive ablations and analyses are provided.

Weaknesses
- The experiments are restricted to one model, Qwen-3 4B, so we cannot know if the method will generalize.
- It seems that retrieval is actually triggered very rarely (Section 5.2.3) around 5%, and I'm not sure if this is enough to actually impact the training process.
- On the same note, I am confused by Section 5.3. which indicates that experience retrieval actually harms performance. Is the paper's claim that incorporating experience stabilizes training over all, but that on any given example, in-context guidance using an experience performs worse?

---

> ### Author Rebuttal · Authors · 2026-03-31
>
> We thank the reviewers for recognizing our effective integration of active experience retrieval into GRPO training, along with consistent benchmark improvements and comprehensive ablations.
>
> **W1: Generalizability**
>
> To validate the generalizability of our method, we conducted additional experiments using Qwen3-8B. The vanilla GRPO baseline obtains an average performance of 31.3%. In contrast, Evolving Rollouts achieves an average of **33.9%**. This performance even surpasses the 32.2% achieved by the significantly larger Qwen3-30B-A3B model. Consistent with our earlier results on Qwen3-4B, these outcomes confirm that the effectiveness of Evolving Rollouts generalises reliably across different models.
>
> **Table 1: Performance comparison between baseline and Evolving Rollouts.**
>
> | Method | GAIA | xBench | HLE | Average |
> | :--- | :--- | :--- | :--- | :--- |
> | Qwen3-8B (Vanilla GRPO) | 40.8% | 44.0% | 9.0% | 31.3% |
> | **Qwen3-8B (Evolving Rollouts)** | **45.6%** | **44.0%** | **12.0%** | **33.9%** |
>
> **W2: Experience Retrieval & Sparsity**
>
> We apologize for any confusion regarding our retrieval mechanics. Our Evolving Rollouts approach differs fundamentally from existing experience-based methods: while prior work relies heavily on retrieval as the main performance driver, our improvements stem from the synergistic co-evolution of the model’s context and parameter spaces during RL training (Figure 2). Regarding the low 5% retrieval rate (Section 5.2.3), our ablation study (Table 2, rows 2 vs. 4) shows that enabling context‑space evolution significantly outperforms the pure parameter‑update baseline. This demonstrates that sparse yet **well‑timed** retrieval is entirely sufficient to guide co‑evolution and deliver consistent gains.
>
> **W3: Selection Bias in Hard Instances**
>
> Concerning the observation in Section 5.3 that retrieval appears to harm performance, we clarify that this reflects an imbalanced comparison stemming from **selection bias**. Because our system uses agentic retrieval, the model autonomously invokes experience *only* when anticipating difficult instances. Therefore, the retrieved examples are inherently much harder. Controlled ablation results (Table 2, rows 1 vs. 2) confirm that the availability of retrieval consistently improves overall system performance.
>
> **Q1: Emergent Adaptive Behavior**
>
> Together with the 5% retrieval rate, our findings show the agent has learned to conduct selective, sparse, and well‑timed retrieval. This adaptive behaviour emerges naturally from GRPO training without any explicit enforcement. We greatly appreciate the reviewer’s comments and will clarify these points in the manuscript to improve transparency.
>
> **Q2: Alternative Embedding Inputs**
>
> While exploring alternative input designs for the embedding model is a valuable direction, it is orthogonal to our core contributions of recovering wasted learning signals. We consider this outside the current scope and leave it for future research.
>
> **Q3: Initial Experience Cold Start Data**
>
> The initial experience cold start data is intentionally distinct from any training data. This design forces the model to learn how to effectively leverage cross-episode experience for entirely different questions, rather than simply memorising the training distribution.

---

> > ### Author Rebuttal · Reviewer_eAMU · 2026-04-02
> >
> > I appreciate the authors' response, as well as additional results with the larger model which show robustness. My concerns are mostly addressed, so I have increased my score.

---

> > > ### Author Response · Authors · 2026-04-02
> > >
> > > Thank you for your positive feedback. Your constructive comments have helped us improve our work.

---

### Official Review · Reviewer_vLg1 · 2026-03-16

**Soundness:** 2
**Presentation:** 2
**Significance:** 3
**Originality:** 2
**Overall Recommendation:** 4
**Confidence:** 3

**Summary:**

This paper proposes an experience-based RL framework to ensure maximum usage of all rollouts of LLMs. By distilling reusable trajectories into internalized behavior, this paper improves the sample efficiency and enhances the final performance across multiple agentic benchmarks. Extensive ablation studies are provided to demonstrate the soundness of the approach.

**Compliance With Llm Reviewing Policy:**

Affirmed.

**Key Questions For Authors:**

NA

**Limitations:**

Please see weaknesses.

**Strengths And Weaknesses:**

## Strengths
- This paper is well-motivated. The discarded learning signals in the GRPO-based algorithm are very wasteful. The core problem is well identified in agentic tasks.
- The central idea is easy to understand: if a rollout is too weak to help via gradients, it may still help via distilled experience.
- Extensive and significant empirical results demonstrate the efficacy of the proposed method.

## Weaknesses
- The overall combination is useful, but the novelty is limited. Techniques like distillation, retrieval-augmented prompting, SFT on generated traces, and RL fine-tuning are not novel and are widely used in recent papers. The main contribution is the integration of these pieces, addressing a practical problem.
- The method introduces additional learning budget through retrieval and experience accumulation. This can increase effective supervision and context length. I suggest that the authors verify whether the baseline could recover some gains simply by reusing the same extra tokens or by storing raw successful traces instead of distilled experiences.

---

> ### Author Rebuttal · Authors · 2026-03-31
>
> We thank the reviewer for recognizing our well-motivated method and its strong empirical performance. We address your specific points below:
>
> **1. Clarification on Novelty: Integration over Individual Components**
>
> While our implementation leverages standard techniques like distillation and RAG, they merely serve as instantiations of our framework's core functions: signal extraction and guidance injection. Our novelty lies not in these individual tools, but in their **synergistic integration to recover and reuse wasted learning signals in group-based RL**. The core innovation is our systematic formulation of this interconnected mechanism to solve signal loss in zero-variance reward groups, supported by rigorous analysis of exactly how these modules interact to make the recovery process work.
>
> **2. New Ablation: Experience Content vs. Context Length**
>
> Following your suggestion, we conducted an ablation study to verify that our gains stem from the actual experience content, rather than simply longer context limits. At each retrieval step, we replaced the extracted experience with previous dialogue turns of an equivalent token length.
>
> As shown in Table 1, our full method significantly outperforms this token-matched baseline. This definitively confirms that the improvements are driven by the **semantic content of the structured experience guidance**, not by the mere injection of additional tokens into the context window.
>
> **Table 1: Performance comparison between the full method and a token-matched ablation.**
>
> | Method | GAIA | xBench | HLE |
> | :--- | :--- | :--- | :--- |
> | Token-Matched Ablation (Previous Turns) | 42.0% | 36.0% | 11.0% |
> | **Full Method (Structured Experience)** | **44.7%** | **36.0%** | **13.0%** |

---

> > ### Author Rebuttal · Reviewer_vLg1 · 2026-04-04
> >
> > Thanks to the authors for the added experiments. I decided to maintain my positive score.

---

> > > ### Author Response · Authors · 2026-04-04
> > >
> > > Thank you for your positive feedback. Your constructive comments have helped us improve our work.

---

### Decision · Program_Chairs · 2026-04-30

**Decision:**

Accept (regular)

**Comment:**

To address the challenge of sparse rewards in agentic RL, this work proposes EVOLVING ROLLOUTS. By leveraging retrieved experiences as in-context guidance, the method makes better use of zero-reward-variance rollouts.

**Strengths**

* Extensive evaluation across three benchmarks with comprehensive ablations
* Simple yet effective approach

**Weaknesses**

* Experiments conducted with only a single backbone model
* Baselines are relatively weak

Overall, all reviewers recognized this as a strong submission, and the authors responded well to reviewer concerns during the discussion period. The paper makes a meaningful contribution that the community will find valuable.